# *Candida albicans* Biofilm Heterogeneity and Tolerance of Clinical Isolates: Implications for Secondary Endodontic Infections

**DOI:** 10.3390/antibiotics8040204

**Published:** 2019-10-30

**Authors:** Om Alkhir Alshanta, Suror Shaban, Christopher J Nile, William McLean, Gordon Ramage

**Affiliations:** Glasgow Dental School, School of Medicine, Dentistry and Nursing, College of Medical, Veterinary and Life Sciences, University of Glasgow, Glasgow G2 3JZ, UKdr.sarah200988@yahoo.com (S.S.); christopher.nile@glasgow.ac.uk (C.J.N.); william.mclean@glasgow.ac.uk (W.M.)

**Keywords:** biofilm, *Candida albicans*, EDTA, NaOCl, tolerance

## Abstract

Aim: Endodontic infections are caused by the invasion of various microorganisms into the root canal system. *Candida albicans* is a biofilm forming yeast and the most prevalent eukaryotic microorganism in endodontic infections. In this study we investigated the ability of *C. albicans* to tolerate treatment with standard endodontic irrigants NaOCl (sodium hypochlorite), ethylenediaminetetraacetic acid (EDTA) and a combination thereof. We hypothesized that biofilm formed from a panel of clinical isolates differentially tolerate disinfectant regimens, and this may have implications for secondary endodontic infections. Methodology: Mature *C. albicans* biofilms were formed from 30 laboratory and oral clinical isolates and treated with either 3% NaOCl, 17% EDTA or a sequential treatment of 3% NaOCl followed by 17% EDTA for 5 min. Biofilms were then washed, media replenished and cells reincubated for an additional 24, 48 and 72 h at 37 °C. Regrowth was quantified using metabolic reduction, electrical impedance, biofilm biomass and microscopy at 0, 24, 48 and 72 h. Results: Microscopic analysis and viability readings revealed a significant initial killing effect by NaOCl, followed by a time dependent significant regrowth of *C. albicans*, but with inter-strain variability. In contrast to NaOCl, there was a continuous reduction in viability after EDTA treatment. Moreover, EDTA significantly inhibited regrowth after NaOCl treatment, though viable cells were still observed. Conclusions: Our results indicate that different *C. albicans* biofilm phenotypes grown in a non-complex surface topography have the potential to differentially tolerate standard endodontic irrigation protocols. This is the first study to report a strain dependent impact on efficacy of endodontic irrigants. Its suggested that within the complex topography of the root canal, a more difficult antimicrobial challenge, that existing endodontic irrigant regimens permit cells to regrow and drive secondary infections.

## 1. Introduction

Fungal infections are generally perceived as being relatively uncommon, yet are reported to affect more than a billion people [1]. This is exacerbated when we consider the alarming global burden of antifungal resistance that we now experience [2]. Many infections we believe to be of bacterial origin are in fact fungal. Indeed, the yeast *Candida albicans* is a ubiquitous human commensal, but with opportunistic tendencies. Its capacity to morphologically transition from yeast to hyphal cells enables it to react dynamically, both in evasion of host immunity and in its ability to readily form biofilm structures that impact increased antifungal resistance [3]. *C. albicans* frequently resides in the oral cavity as a biofilm forming microorganism, interacting with other oral microbiota and the host. When we consider root canal infections, pathogenic yeasts have also been isolated from teeth associated with primary apical periodontitis and post-treatment disease [4].

Dental pulp is generally a sterile tissue containing nerves and vascular tissues that are connected to the surrounding periradicular tissues, though recent studies appear to suggest the presence of bacterial DNA in the pulp of pristine healthy teeth [5]. The pulp is protected from the oral microbiome by layers of mineralised tissues (dentin, enamel and cementum). Breach to these tissues, as a result of dental caries, tooth fractures or cracks, can expose the dentin and provide routes for oral microbiota to ingress towards the pulp [6]. Another route of pulp infection is the direct invasion of microorganisms to the pulp during root canal treatment during substandard clinical procedures. *C. albicans* is the most frequently isolated species in endodontic infections [7,8]. The prevalence of yeast in persistent infections is higher than that in primary infections [4]. However, two recent systematic reviews found that *C. albicans* prevalence does vary significantly between primary and secondary endodontic infections [7,8]. In this environment *C. albicans* can colonise dentinal walls and penetrate into dentinal tubules [9,10]. The route of fungal invasion to the root canals is likely to be through cracks or via leaking restoration, though the diameter of hyphae, which is 1.9–2.6 µm [11], cannot exclude the possibility of dentinal tubules invasion especially in deep caries lesions. Here, candida yeasts are capable of coalescing with one another in the form of biofilm, which is a multicellular community of yeast and hyphal forms encased in polymeric glue. The biofilm lifestyle enhances the ability of the cells to withstand host and antifungal factors, ultimately contributing to its persistence within the root canal. This phenotype, along with the complexity of root canal system, particularly in the apical third of root canals, are thought to be the major causes of treatment-resistant apical periodontitis.

Root canal treatment (RCT) is the treatment of choice for these endodontic infections. RCT aims to: (1) eliminate microorganisms from the root canal system to a level that promotes the healing of periradicular tissues, and (2) provide a three-dimensional hermetic seal to prevent reinfection. Different hand and rotary instruments are used during RCT in order to mechanically debride the biofilm on the root canal walls. Nevertheless, the cross-sectional root canal configuration can pose a challenge to adequate debridement, as these instruments tend to leave untouched recesses in oval canals [12,13]. This instrumentation can however create a smear layer that covers the root canal walls, which consists of organic (pulp tissue remnants) and inorganic (dentin chips) tissues [14]. This layer might act as a protective barrier to encase biofilms formed on root canal walls [15] and might also encourage the adherence of microorganisms, such as *C. albicans* [16]. Furthermore, it may compromise the quality of root canal sealing [17]. Therefore, the use of chemical irrigating solutions to maximise root canal debridement and removal of the smear layer is vital for successful RCT, which includes ethylenediaminetetraacetic acid (EDTA), though not universally [18]. EDTA acts mainly as an adjunct irrigant to remove smear layer, though an alternative effect is the inhibition of filamentation through chelation of necessary divalent cations in the pathogenic yeast *C. albicans* [19], a structural element strongly associated with robust biofilm formation.

Recent work from our group has highlighted the clinical relevance of biofilm heterogeneity between isolates of *C. albicans* from the oral cavity and other clinical sites [20]. The overriding message from this review is that with respect to particular groups of microorganisms, they cannot all be considered as a single entity and “one size does not fit all” when it comes to treatment regimens. We have shown that the greater capacity to form biofilm is linked to enhanced virulence and increased resistance to antifungals [21]. Further studies of this yeast, alongside the emerging yeast pathogen *C. auris*, have demonstrated that biofilm-mediated patterns of resistance also exist for both antiseptics and disinfectants, including sodium hypochlorite [22]. Given that sodium hypochlorite (NaOCl) and EDTA are principal components of the management of endodontic infections [23], we hypothesised that root canal treatment failure involving *C. albicans* may be driven by biofilm heterogeneity resulting in an enhanced ability to withstand treatment and regrow within root canals. We therefore aimed to assess how different biofilm phenotypes responded in vitro to NaOCl and EDTA treatments.

## 2. Material and Methods

### 2.1. Microbial Growth Conditions and Standardisation

*C. albicans* laboratory strains SC5314 [24] and 3153A [25], and 28 clinical isolates obtained from an oral rinse from patients attending restorative clinics at Glasgow Dental Hospital and School for routine dental care, as previously described [26]. Antifungal susceptibility profiles and proteolytic activity of these strains have been described previously [27,28]. Notably, these studies showed that these isolates (saliva derived) were highly sensitive planktonically to azoles, polyenes and echinocandins, whereas in sessile mode they were resistant to only azoles [28]. Moreover, proteolytic assessment of these isolates showed that those of a higher biofilm forming capacity were more proteolytic [27]. The strains were subcultured on Sabouraud’s dextrose agar [SAB (Sigma–Aldrich, Dorset, UK)] plates and maintained at 30 °C for 48 hours. An overnight culture was prepared in yeast peptone dextrose (YPD (Sigma-Aldrich, Dorset, UK)) and incubated at 30 °C at 120 rpm in an orbital shaker (IKA KS 4000 i control, Berlin, Germany). After 18 h, the yeast cells were pelleted by centrifugation, washed twice with phosphate buffered saline (PBS (Sigma–Aldrich, Dorset, UK)) and counted using a haemocytometer.

### 2.2. Biofilm Development and Biofilm Forming Ability Characterisation

*C. albicans* strains were standardised to the desired cellular density of 1 × 10^6^ cells/mL into Roswell Park Memorial Institute (RPMI)-1640 (Sigma–Aldrich, Dorset, UK) medium and the biofilms formed onto pre-sterilised, polystyrene, 96-well flat-bottom microtiter plates (Corning Incorporated, Corning, NY, USA) as previously described [29]. The plates were incubated at 37 °C for 24 h. After incubation, the biofilms were washed with PBS to remove the loosely attached cells and the biofilm biomass of all strains was quantified using the crystal violet (CV) assay, as previously described [30]. The biomass was then quantified spectrophotometrically by reading absorbance at 570 nm using microtiter plate reader (FLUOStar Omega, BMG Labtech, Aylesbury, UK).

The growth profiles of the selected test strains were also assessed using an xCELLigence system (ACEA, Biosciences Inc., San Diego, California, USA). Briefly, this system is a real-time cell analysis instrument that uses patented microtitre plates (E-plates^®^) that contain gold biosensors embedded in the bottom of each well. Quantifiable data was expressed as cell index, which represents the cell impedance when the cells adhere, or associated matrix, to the electrodes [31]. These biosensors continuously and non-invasively monitor changes in cell number, cell size, cell-substrate attachment quality, and extrapolymeric substance (EPS) production. *C. albicans* (1 × 10^6^ cells/mL) in RPMI-1640 was seeded into 96-well E-plates^®^ (ACEA, Biosciences Inc.) and the plates incubated for 24 h at 37 °C with readings taken every 30 min.

### 2.3. Planktonic Minimum Inhibitory Concentration (MIC)

MIC was determined using a broth microdilution method according to the M27-A3 standard for fungi [32] (CLSI). Briefly, cells were adjusted to the desired density of 2 × 10^4^ cells/mL in Roswell Park Memorial Institute (RPMI) media (Sigma–Aldrich, Dorset, UK). A series of two-fold dilutions of 3% (30,000 ppm) NaOCl (Parcan; Septodont, Saint-Maur-des-Fosses, France) and 17% (EDTA [ENDO-SOLution, Stalowa Wola, Poland]) were performed using 96-well round-bottom microtiter plates (Corning Incorporated, Corning, NY, USA). The plates were at 37 °C. After 24 h, the MIC concentration was determined as the lowest concentration that prevents visible growth.

### 2.4. Biofilm Treatment and Regrowth Assessment

After biofilm growth, they were washed with PBS and treated with 3% NaOCl, 17% EDTA for 5 min, or sequentially treated with NaOCl followed by EDTA for 5 min each. Untreated controls were used for comparison. The effect of NaOCl was deactivated with 5% sodium thiosulfate for 10 min (Fisher Chemicals, London, UK) and EDTA was deactivated using Dey Engley Neutralising broth (Sigma-Aldrich, London, UK) for 15 min. Sodium thiosulfate and Dey Engley Neutralising broth were also applied to untreated controls to normalise results. The viability of the biofilms was quantified immediately after treatment using XTT metabolic activity assay (Sigma-Aldrich, UK) and CV as described above. The absorbance was measured spectrofluorometrically using a plate reader (FLUOStar Omega, BMG Labtech, Aylesbury, UK) at a wavelength of 490 nm for XTT and 570 nm for CV according to the manufacturer’s recommendation. To further explore the effect of this treatment, post-treatment, the microtitre plates were washed with PBS and fresh RPMI-1640 media replenished. The plates were re-incubated at 37 °C for 24, 48 and 72 h and XTT viability and CV assessment performed, as described above. To validate this approach, we mirrored this experimental procedure using the xCELLigence system. Instead of static endpoint measurements, we were able to monitor biofilm regrowth in real-time. Briefly, following the initial treatment, deactivation and washing steps, the wells were replaced with RPMI and the entire plate sonicated for 5 min to remove retained adherent biofilm cells. The sonicated cells were then transferred into 96-well plastic E-plates, and the plates re-incubated at 37 °C for 72 h, with cell impedance quantified every 30 min. In parallel, plates were imaged using EVOS FL Cell Imaging system (Thermo Fisher Scientific, Waltham, MA, USA).

### 2.5. Statistical Analysis

Viability and biomass data of *C. albicans* BC023 and BC146 were chosen to represent the low biofilm forming (LBF) and high biofilm forming (HBF) strains respectively. GraphPad Prism (version 7.0 d) was used for creating graphs and statistical analysis. Data was tested for normal distribution using D’Agostino-pearson omnibus normality test and were plotted as log were required. Student *t*-test was used to compare LBH and HBF viability. One-way ANOVA with Tukey’s multiple comparison test was used to statistically compare biofilm viability after NaOCl treatment for each isolate over time while A Kruskal–Wallis test with Dunn’s multiple comparison test was used to compare viability following the three treatment conditions.

## 3. Results

### 3.1. Oral Candida albicans differentially form biofilms and are NaOCl sensitive

Initially, biofilm biomass was assessed for the 28 *C. albicans* clinical isolates (Figure 1). Isolates showed different biofilm forming abilities with the biofilm biomass for 18 isolates ranges between 0.4–1.0 mean optical density. The five lowest biofilm forming isolates (LBFs) and the five highest biofilm formers (HBFs) were subsequently selected to test their susceptibility to NaOCl and EDTA treatment. In terms of growth profiles, BC023 (the lowest biofilm forming clinical isolate), BC146 (the highest biofilm forming clinical isolate), 3153A (low biofilm forming laboratory strain) and SC5314 (high biofilm forming laboratory strain) were used as controls. Growth profiles of BC023 and 3153A (Figure 2a) were clearly different from those of BC146 and SC5314 (Figure 2b). LBFs showed less irregularities before both reach saturation phase when the E-plate electrodes become entirely covered with *Candida* cells. This difference suggests the potential of xCELLigence system to differentiate LBF and HBF isolates based on their growth profiles. NaOCl MIC for all LBFs and HBFs was 0.093% (930 ppm) and 0.13% for EDTA.

### 3.2. Oral Candida albicans persistsfFollowing NaOCl treatment

Following NaOCl application, cell viability expressed in percentage in relation to untreated controls was assessed (Figure 3). In spite of the initial killing effect of NaOCl (78.7%, 98.3% reduction in viability for LBF and HBF, respectively), both isolates were able to regrow over time. LBF and HBF showed comparable regrowth profiles. However, LBF revealed higher immediate post-treatment viability compared to HBF (*p* < 0.05). In terms of post-treatment regrowth, the viability increased significantly between 0 and 72 h for both LBF (*p* < 0.0001) and HBF (*p* < 0.01) with maximum viability at 48 h (*p* < 0.0001). No significant difference was observed between viability at 48 and 72 h for both isolates. Data from xCELLigence system confirm the above-mentioned results for HBF (Figure 4b), but not for LBF (Figure 4a).

### 3.3. Oral Candida albicans regrowth after NaOCl treatment is significantly inhibited by EDTA

As showed in the previous data, *C. albicans* persist following NaOCl treatment and have the potential to regrow. In terms of EDTA treatment, the metabolic activity of LBF (Figure 5a) and HBF (Figure 5b) was initially decreased by 82%, 97%, respectively, when compared to untreated biofilms. Again, LBF showed significantly higher immediate post-treatment tolerance to EDTA than HBF (*p* < 0.001). In contrast to NaOCl, the metabolic activity following EDTA treatment showed a continued reduction between 0 and 72 h for both isolates. Similar trends were observed for NaOCl + EDTA treatment of LBF, but not for HBF. The metabolic activity is significantly lower in EDTA and NaOCl + EDTA in comparison to NaOCl alone for both isolates at 48 (*p* < 0.0001) and 72 h (*p* < 0.05, *p* < 0.01 for LBF and *p* < 0.0001 for HBF). Finally, there was not a significant difference between EDTA and NaOCl + EDTA at all time points.

Increased viability over time with NaOCl treated biofilms was not matched with an increase in biofilm biomass for both isolates. Furthermore, EDTA treated HBF shows significantly higher biomass in comparison to NaOCl and NaOCl + EDTA at all time points (Figure 6b). At 72 h, NaOCl + EDTA is not superior in decreasing biofilm biomass compared to NaOCl alone for HBF, but it is more effective in decreasing biomass compared with NaOCl treatment in LBF (*p* < 0.001) (Figure 6a).

Microscopic examination is consistent with these data (Figure 7). There are visibly evident cells for both isolates at 0 h in the all treatment conditions (Figure 7a,b). However, the visible quantity of cells increased significantly over time in NaOCl treated biofilms for both isolates. In EDTA treatment, the number of cells remained relatively stable over time in both isolates, with maintaining biofilm architecture in HBF. Finally, NaOCl + EDTA showed significantly less cells at 24, 48 and 72 h compared with NaOCl alone and EDTA alone.

## 4. Discussion

*C. albicans* biofilm infections are heterogeneous, and it has been shown that depending on the phenotype of a specific clinical isolate, this may have a profound impact on patient outcomes [33]. Therefore, we sought to test whether these characteristics played a role in a dentally relevant scenario. Root canal treatment is the treatment of choice for endodontic infections. The use of chemical irrigating solutions to disinfect the root canal and to remove the smear layer is vital for success. An ideal irrigant should provide a flushing action, have a broad spectrum antimicrobial activity, be able to inactivate bacterial endotoxins, have the ability to dissolve the necrotic organic tissues, remove the smear layer, have no adverse effect on dentin and be biocompatible with the periradicular tissues [34]. However, no single irrigant possesses all of these characteristics. Here we modelled the in vitro ability of standard endodontic irrigants to manage *C. albicans* biofilms, demonstrating a flaw in the capacity of NaOCl and EDTA to effectively eradicate the biofilm.

The present study has demonstrated a significant variation between candidal phenotypes, with LBF (mainly yeasts) and HBF (hyphal forms) differing in their susceptibility to NaOCl and EDTA. Although HBF would be expected to show higher resistance to these irrigants attributed to its higher biofilm forming ability, LBF showed greater tolerance in comparison to untreated control than HBF, possibly due to the capacity of budding yeast cells to disperse. Radcliffe et al (2004) previously examined the resistance of two strains of *C. albicans* to NaOCl of concentrations ranges from 0.5% to 5.25% and found that NaOCl as low as 0.5% lowered the colony forming units (CFU) to below the detection limit for the tested strains [35]. No difference was noted between the response of the tested strains. However, this study examined treatment of planktonic cells rather than biofilm. As highlighted previously, it is becoming clear that the heterogeneity of microbial isolates requires attention when considering treatment strategies, it is no longer appropriate to assume that because a protocol is effective for one clinical isolate that it will be equally effective for another. We are now in the era of personalised antimicrobial therapy; although this may not be directly relevant in endodontic treatment, it does highlight the potential need for fine tuning of irrigation protocols that can take into account differential sensitivities.

In an attempt to enhance the sensitivity of the analysis, we employed an xCELLigence system that relies on electric impedance. Not previously used in either fungal or endodontic studies, the system provided the potential to detect early phases of regrowth. It allowed determination of biofilm growth kinetics in untreated samples, however, was less informative following NaOCl treatment. Due to the degradation of the gold electrodes used in the E-plates, it was necessary to grow and treat biofilms in conventional polystyrene, 96-well flat- bottom microtiter plates then sonicate and transfer the persistent cells to xCELLigence system E-plates. There is a possibility that cells were lost during sonication or transfer which may account for the poorly detectable regrowth in some LBF strains. Nonetheless, the novelty of this real-time biofilm monitoring system enabled us to assess biofilm regrowth in real-time for several of these treated isolates.

NaOCl is the most widely used endodontic irrigant due its potent and wide spectrum antimicrobial properties and organic tissue dissolving ability, although it may be associated with several risks such as tissue toxicity, emphysema, allergy and undesirable taste and smell [36]. Various studies investigated the effect of different NaOCl concentrations on *C. albicans* at a variety of clinically relevant treatment time points and concentrations. Previously, it was shown that NaOCl has the highest efficacy against *C. albicans* [37]. They showed that NaOCl (0.5% and 5%) completely killed all *C. albicans* cells in 30 sec, with all the tested *C. albicans* strains showing similar susceptibility. However, our results show that with 3% NaOCl application for 5 min on biofilms, some *C. albicans* cells were still viable and are able to regrow. Waltimo et al (1999) tested the NaOCl against planktonic *Candida* cells, whereas 24 h biofilm was used in the present study which may explain the different results [37]. This is particularly true when considering that biofilms are believed to be significantly more tolerant to antimicrobials than planktonic free cells [38,39]. Given the nature of endodontic infections that develop over a period of time, and the fact that *C. albicans* has a great affinity for the collagen of the dentine which promotes yeast adhesion [10], it is more likely that *C. albicans* will occur in a biofilm rather than planktonic cells in endodontic infections.

Recently, similar findings to our results were reported, where viable candidal cells were retrieved from root canals following 3% treatment and incubation with growth media for 72 h [40]. However, they tested the effect of NaOCl on *C. albicans* grown in root canals of previously healthy teeth, therefore, it is more likely that viable cells would be retrieved due to the complexity and topography of root canals and the presence of smear layer which may impede the action of NaOCl. Whereas, in the present study biofilms were grown and treated on an uncomplicated 96-well flat-bottom microtitre plate. It is widely believed that persistence of endodontic infections is mainly due to complexity of root canal anatomy which impedes the disinfection of inaccessible areas [12,13]. In addition, the buffering action of dentine and the small volume of irrigant within root canals are believed to decrease the killing efficiency of NaOCl [41,42]. However, our results challenge this concept. We were able to demonstrate that even though *C. albicans* biofilms were entirely exposed to a large volume of NaOCl in the absence of dentine buffering effect, eradication of cells was unachievable.

EDTA is recommended as a part of endodontic irrigation protocols. Its main action is to remove the smear layer to enhance the penetration of other irrigants due to its chelating action (it chelates calcium ions from inorganic tissues). EDTA was reported, by many studies, to have a high antifungal efficacy at concentration as low as 0.625 mg/L [43]. They suggest that EDTA antifungal activity is attributed to its ability to decrease the metabolic activity by extracting calcium ions from *C. albicans* cell wall and the medium and its anti-colonization action by reducing adhesion properties of *C. albicans*. Calcium ions have a crucial role on the morphogenesis and pathogenesis of *C. albicans* by affecting adherence and growth [19]. Interestingly, the metabolic activity of EDTA treated HBF was significantly lower while the biofilm biomass of the same biofilms was significantly higher in comparison to untreated biofilms. Furthermore, biofilm architecture of the same biofilms was maintained when viewed microscopically. EDTA inhibits the metabolic activity of *Candida* cells, which may explain the low XTT readings while maintaining high biomass. Previous research reported that EDTA inhibits biofilm formation of *C. albicans*, but minimally affects pre-formed biofilms [19]. It was also reported that growth and biofilm formation can be restored if EDTA is removed and fresh media is added. However, our present findings showed a significant reduction in viability of EDTA treated pre-formed biofilms. Furthermore, there was a continuous reduction in viability over time rather than regrowth. This variation can be explained by the significant difference in EDTA concentration. A clinically relevant EDTA concentration of 17% (equivalent to 581 mM) was used in this study, whereas the highest EDTA concentration used by Ramage and colleagues was 250 mM [19].

Contrary to the results at our baseline time point, Sen and colleagues (2000) showed that EDTA was superior to NaOCl in its antifungal effect [43]. In our study, there was no difference between the two disinfectants on LBF, but there was a higher antifungal effect observed with NaOCl on HBF. This group also used agar diffusion tests to compare the two, whilst we used biofilms and the XTT reduction assay to assess antifungal activity which may account for the difference in the results [9]. Our findings highlight the importance of including EDTA in irrigation protocols not only for smear layer removal, but also for its antifungal properties and its ability to inhibit candidal regrowth after treatment with NaOCl. It is important to note that our studies did not explore the effect of irrigant activation, which is an integral part of endodontic disinfection protocols and could potentially, in combination with NaOCl and EDTA, eliminate *C. albicans* and prevent its regrowth. Furthermore, *C. albicans* coexist with multiple species and are known for their ability to form biofilms with and without other bacterial species (mono-microbial and poly-microbial biofilms), such as *Enterococcus faecalis* and *Staphylococcus aureus* [44]. As a result, future investigation should explore whether interkingdom interactions of *C. albicans* with bacteria could complicate root canal treatment and provide protection to one another ultimately leading to persistence.

In conclusion, *C. albicans* clinical isolates have variable biofilm forming abilities, which differentially tolerate endodontic irrigants. These isolates were able to persist after 3% NaOCl treatment and regrow to levels that are comparable with untreated biofilms, with more regrowth potential observed for yeast cell phenotypes. EDTA significantly inhibits the persistence of NaOCl treated biofilms, though not entirely, indicating possibilities for secondary endodontic infections.

## 5. Conclusions

This first of its kind study reports a strain dependent impact on efficacy of endodontic irrigants. The *in vitro* data suggests that existing endodontic irrigant regimens are not 100% efficient, and that biofilm tolerant cells are permitted to repopulate their immediate physical surroundings. The clinical implications for these observations is the potential for a secondary infections driven by initial ineffectual endodontic treatment. Clinicians conducting endodontic therapies should be aware of this microbiological phenomenon, and support the development of alternative augmentative treatments. 

## Figures and Tables

**Figure 1 antibiotics-08-00204-f001:**
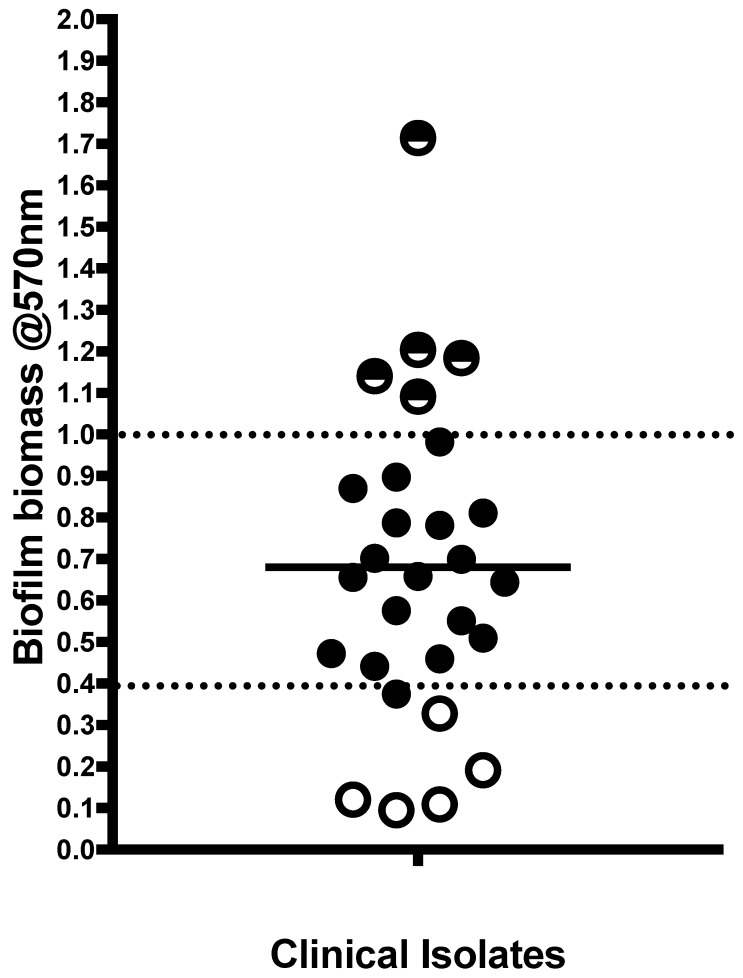
Quantification of biofilm biomass from clinical isolates. low biofilm forming (LBF) and high biofilm forming (HBF) identified by horizontal lower and higher dotted lines. Those isolates with an open circle or half dark/white circle represent the five LBF and HBF isolates selected for downstream analysis, respectively.

**Figure 2 antibiotics-08-00204-f002:**
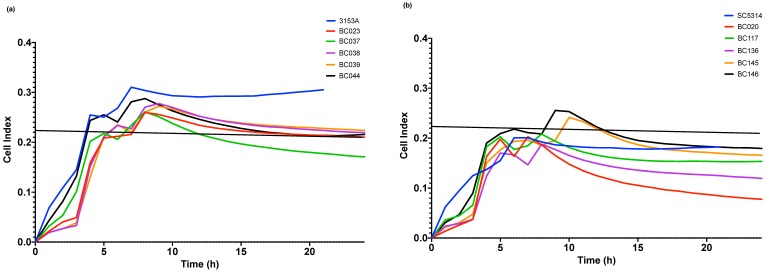
Growth profile as shown by xCELLigence system. Data represent the mean of normalized cell index values taken every 30 min over 24 h of (**a**) *C. albicans* 3153A and five LBFs (**b**) *C. albicans* SC5314 and five HBFs.

**Figure 3 antibiotics-08-00204-f003:**
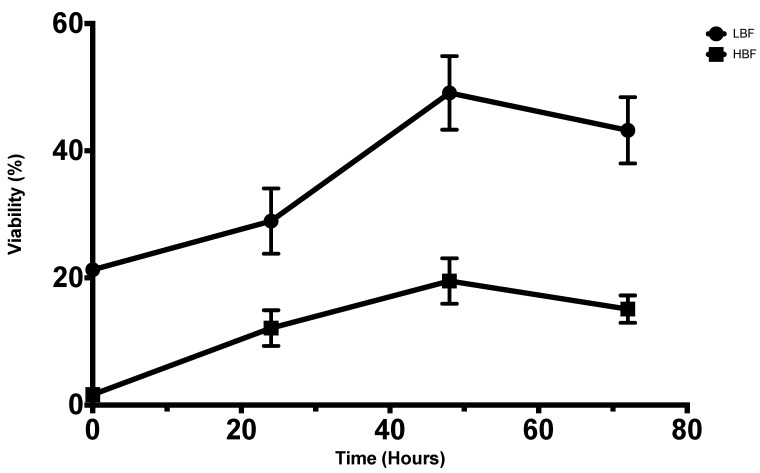
Viability percentage of biofilms in relation to untreated biofilms after treatment with NaOCl and reincubation with fresh Roswell Park Memorial Institute (RPMI) media. Immediate post-treatment (0), 24, 48 and 72 h for LBF (upper line) and HBF (lower line). Values represent the mean with standard error of the mean (SEM) from three independent experiments.

**Figure 4 antibiotics-08-00204-f004:**
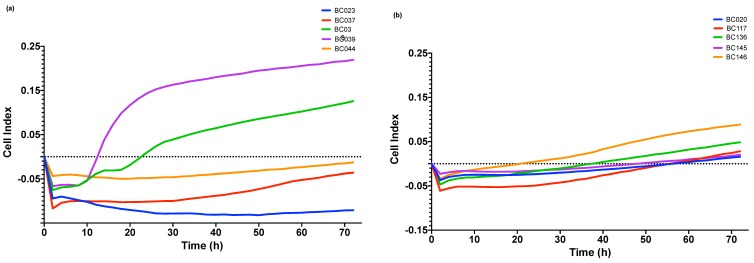
Regrowth from 0 to 72 h as detected by xCELLigence system and presented as normalized cell index for (**a**) five LBF (**b**) five HBF.

**Figure 5 antibiotics-08-00204-f005:**
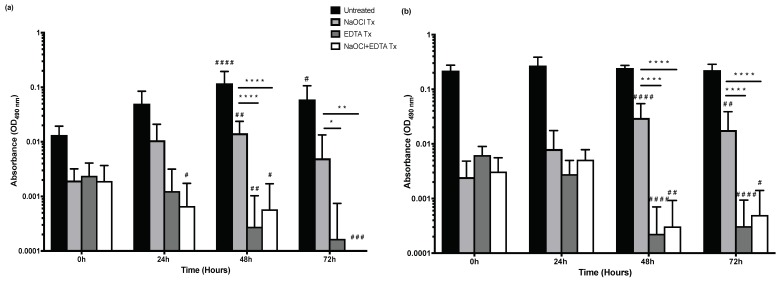
The effect of the three treatment conditions: NaOCl, ethylenediaminetetraacetic acid (EDTA) and NaOCl + EDTA on pre-formed 24 h biofilms viability in relation to untreated biofilms. Biofilms were treated for 5 min and reincubated with fresh RPMI. XTT readings were taken at 0, 24, 48 72 h. Values were plotted as log_10_ on the Y axis. (**a**) absorbance values for LBF (**b**) absorbance values for HBF. Statistical significance between the three treatment conditions at 48, and 72 h was presented as * *p* < 0.05, ** *p* < 0.01 and **** *p* < 0.0001. Whereas the significant difference in relation to untreated biofilms of the same treatment condition over time was presented as ^#^
*p* < 0.05, ^##^
*p* < 0.01, ^###^
*p* < 0.001 and ^####^
*p* < 0.0001.

**Figure 6 antibiotics-08-00204-f006:**
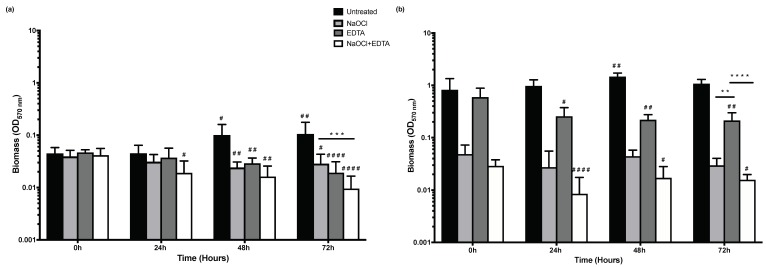
The effect of the three treatment conditions: NaOCl, EDTA and NaOCl + EDTA on pre-formed 24 h biofilms biomass in relation to untreated biofilms. Crystal violet (CV) readings were taken at 0, 24, 48 and 72 h. Values were plotted as log_10_ on the Y axis. (**a**) Biofilm biomass for LBF (**b**) Biofilm biomass for HBF. Statistical significance between the three treatment conditions at 72 h was presented as ** *p* < 0.01, *** *p* < 0.001 and **** *p* < 0.0001. Whereas the significant difference in relation to untreated biofilms of the same treatment condition over time was presented as ^#^
*p* < 0.05, ^##^
*p* < 0.01 and ^####^
*p* < 0.0001.

**Figure 7 antibiotics-08-00204-f007:**
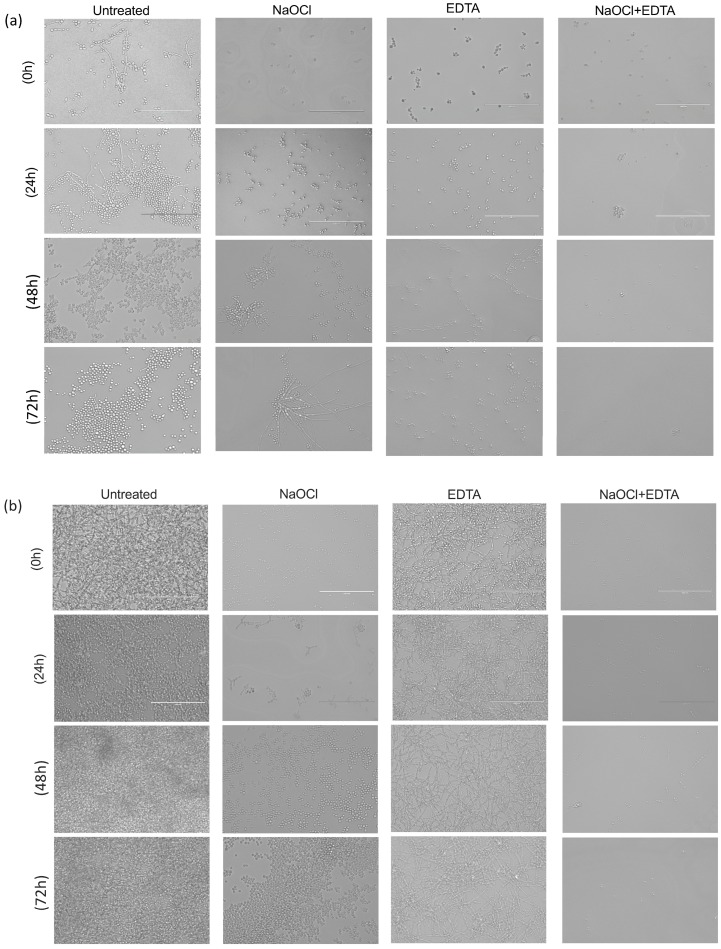
Microscopic examination of biofilms at four conditions: untreated, NaOCl, EDTA and NaOCl + EDTA treated biofilms. It demonstrates the biofilms at four time points, 0, 24, 48 and 72 h. (**a**) LBF (**b**) HBF. Bar is 40 μM for all panels.

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
