# Peer review of "Candida albicans Biofilm Heterogeneity and Tolerance of Clinical Isolates: Implications for Secondary Endodontic Infections"

_antibiotics, 2019, doi:10.3390/antibiotics8040204_

Round 1

Reviewer 1 Report

The paper is well documented and the topic fits within the Journal aim and scope.

Just some minor concerns should be addressed by the authors in order to have a final more strong paper. 

Introduction and discussion section are weak. Please make a deep revision of the literature about endodontics and dental pathology in order to compare the data of the presented work with the others just published in the recent literature. Sometimes the presence and prevalence of the Candida can be related to the antibiotics assumption and not only related with the endodontic treatements. Some sample as follows.

Cervino, G.; et al. Antibiotic Prophylaxis on Third Molar Extraction: Systematic Review of Recent Data. Antibiotics 2019, 8, 53

Author Response

We thank reviewer 1 for their helpful comments. The weakness in introduction and has now been addressed and new sections included within the revised text to strengthen the endodontic literature. There are numerous mentions of endodontics in the discussion, so we felt it was not necessary to add further.`

“Root canal treatment (RCT) is the treatment of choice for these endodontic infections. RCT aims to: (1) eliminate microorganisms from the root canal system to a level that promote the healing of periradicular tissues, and (2) provide a three-dimensional hermetic seal to prevent reinfection. Different hand and rotary instruments are used, during RCT in order to mechanically debride the biofilm on the root canal walls. Nevertheless, the cross-sectional root canal configuration can pose a challenge to adequate debridement, as these instruments tend to leave untouched recesses in oval canals (12, 13). This instrumentation can however create a smear layer that covers the root canal walls, which consists of organic (pulp tissue remnants) and inorganic (dentin chips) tissues (14). This layer might act as a protective barrier to encase biofilms formed on root canal walls (15) and might also encourage the adherence of microorganisms, such as C. albicans (16). Furthermore, it may compromise the quality of root canal sealing (17). Therefore, the use of chemical irrigating solutions to maximise root canal debridement and removal of the smear layer is vital for successful RCT, which includes ethylenediaminetetraacetic acid (EDTA), though not universally (18). EDTA acts mainly as an adjunct irrigant to remove smear layer, though an alternative effect is the inhibition of filamentation through chelation of necessary divalent cations in the pathogenic yeast C. albicans (19), a structural element strongly associated with robust biofilm formation.”

In terms of the possible relation between presence and prevalence of the Candida and antibiotics, the use of antibiotics in endodontics is not strictly relevant. We thank the reviewer for the suggested reading, but it does not seem appropriate for the direction of study. Firstly, according to the guidelines on the use of antibiotics in endodontic infections, systemic antibiotics are only indicated in few limited cases, such as immunocompromised patients, endodontic infections with systematic involvement (periapical abscesses). Therefore, it is unlikely that candida growth or tolerance inside root canals is a consequence of antibiotic use – its more likely these appear during secondary infection when saliva with yeasts present enter the canal. Secondly, systemic antibiotics cannot reach root canals as there is no blood circulation due to the death and degeneration of pulp tissue.

Reviewer 2 Report

In this study, the authors showed that there is a strain dependent impact on efficacy of endodontic irrigants. For that the authors used different C. albicans clinical isolates. 

Some points that need to be taken into account:

#1 What are the pathogenic characteristics of the isolates (e.g. antifungal susceptibility profiles, etc)?                                                                           #2 The MIC was determined in MHB instead of RPMI medium;                          #3 Microscopic examination of biofilms. There are some inconsistencies in the images from the same panel (e.g Fig 7a for the untreated control at 0-48h we clear can observed hypha formation while at 72h this is not observed).

Author Response

We thank you for your relevant and insightful comments.

#1 What are the pathogenic characteristics of the isolates (e.g. antifungal susceptibility profiles, etc)?                                                                          

Antifungal susceptibility of the used clinical isolates has been previously determined in a much earlier studies where the isolates were derived through clinical collection. Ramage et al. 2011. Oral Surg Oral Med Oral Pathol Oral Radiol Endod. 2011 Apr;111(4):456-60. doi: 10.1016/j.tripleo.2010.10.043.Commercial mouthwashes are more effective than azole antifungals against Candida albicans biofilms in vitro.

These studies showed that these isolates (saliva derived) were highly sensitive planktonically to azoles, polyenes and echinocandins, whereas in sessile mode they were resistant to azoles only (a consequence of their fungistatic mode of growth, i.e. intact biofilms stop growing and expanding).

We have also investigated protease activity in these isolates – Ramage et al. 2012. Mycopathologia. Jul;174(1):11-19. doi: 10.1007/s11046-012-9522-2. In vitro Candida albicans biofilm induced proteinase activity and SAP8 expression correlates with in vivo denture stomatitis severity.

This study showed that protease activity was more preferential in higher biofilm forming isolates, but plays no bearing on this study.

In terms of the broth microdilution testing, this was an inadvertent mistake. It was performed in RPMI instead of MHB. To ensure we are not in doubt, the MIC was re-determined using RPMI and shown not to differ from our original testing. The text was altered and highlighted.

The inconsistencies of the microscopic images resulted from the fact that identical (but not the same) plates and wells were imaged at different time points. For each candida isolate, biofilm was grown, treated, imaged and assessed using XTT and CV using 4 identical plates, each plate represents a single time point. Furthermore, candida isolate in Fig 7a is a low biofilm forming isolates where the biofilm is mainly formed by cells with occasional hyphae.   

Reviewer 3 Report

In this work, the authors studied the ability of C. albicans to tolerate treatment with standard endodontic irrigants NaOCl, EDTA and a combination. This interesting study may provide some implications and explanations for secondary endodontic infections. The work is well done and the presentation is clear. The conclusions are also very useful for future diagnose and therapeutic. I just have three minor comments.

What’s the mechanism of EDTA as a principal component of the management of endodontic infections, especially on the formation of biofilm? It would be clearer if give a brief description in the introduction What are the advantages of the examination of planktonic cell treatment instead of biofilm in this study? Please give more details about the xCELLigence system.

Author Response

We thank you for your relevant and insightful comments.

The clinical and biological mechanism of EDTA as a component of root canal treatment has been added (highlighted text).

Therefore, the use of chemical irrigating solutions to maximise root canal debridement and removal of the smear layer is vital for successful RCT, which includes ethylenediaminetetraacetic acid (EDTA), though not universally (18). EDTA acts mainly as an adjunct irrigant to remove smear layer, though an alternative effect is the inhibition of filamentation through chelation of necessary divalent cations in the pathogenic yeast C. albicans (19), a structural element strongly associated with robust biofilm formation.

We apologise for the confusion, this study investigated the effect of treatment on biofilm, not planktonic cells (planktonic data was included for comparison). In endodontic infection, microorganisms enter, colonize and form biofilm inside root canal before they can reach the periapical tissue and cause symptoms, thereby, treatment can be started. It very unlikely that microorganisms will present in a planktonic form in established endodontic infections.

We have added the extra highlighted text for the xCELLigence system:

The growth profiles of the selected test strains were also assessed using an xCELLigence system (ACEA, Biosciences Inc.). Briefly, this system is a real-time cell analysis instrument that uses patented microtitre plates (E-plates®) that contain gold biosensors embedded in the bottom of each well. Quantifiable data was expressed as cell index, which represents the cell impedance when the cells adhere, or associated matrix, to the electrodes (29).

The mechanism of EDTA as a component of root canal treatment has been briefly discussed in page 10 line 296-298 and page 11 line 299 (more information on what is smear layer and why it’s important to remove it is available on the paragraph you will add). The effect of EDTA on candida biofilm formation was discussed in page 11 line 299-303) we cited the literature demonstrates that EDTA inhibits colonization and adhesion of candida and therefore biofilm formation. We also cited the work from our group that reports candida biofilm formation inhibition by EDTA in page 11 line 307-309.

In terms of the xCELLigence system, we have added the following highlighted section in the introduction:

Briefly, this system is a real-time cell analysis instrument that uses patented microtitre plates (E-plates®) that contain gold biosensors embedded in the bottom of each well. Quantifiable data was expressed as cell index, which represents the cell impedance when the cells adhere, or associated matrix, to the electrodes (21).

Round 2

Reviewer 2 Report

I would like to see in the main text of the manuscript similar comments about the clinical strains:

"Antifungal susceptibility of the used clinical isolates has been previously determined in a much earlier studies where the isolates were derived through clinical collection. Ramage et al. 2011. Oral Surg Oral Med Oral Pathol Oral Radiol Endod. 2011 Apr;111(4):456-60. doi: 10.1016/j.tripleo.2010.10.043.Commercial mouthwashes are more effective than azole antifungals against Candida albicans biofilms in vitro.

These studies showed that these isolates (saliva derived) were highly sensitive planktonically to azoles, polyenes and echinocandins, whereas in sessile mode they were resistant to azoles only (a consequence of their fungistatic mode of growth, i.e. intact biofilms stop growing and expanding).

We have also investigated protease activity in these isolates – Ramage et al. 2012. Mycopathologia. Jul;174(1):11-19. doi: 10.1007/s11046-012-9522-2. In vitro Candida albicans biofilm induced proteinase activity and SAP8 expression correlates with in vivo denture stomatitis severity.

This study showed that protease activity was more preferential in higher biofilm forming isolates, but plays no bearing on this study".

Author Response

Thank you for these further comments, and we apologise for not adding these details into the manuscript at the time. These rebuttal points have been integrated within the relevant methodological sections.

"Antifungal susceptibility profiles and proteolytic activity of these strains have been described previously (27, 28). Notably, these studies showed that these isolates (saliva derived) were highly sensitive planktonically to azoles, polyenes and echinocandins, whereas in sessile mode they were resistant to only azoles (28). Moreover, proteolytic assessment of these isolates showed that those of a higher biofilm forming capacity were more proteolytic (27)."